# Design and Optimization of Tool-Embedded Thin-Film Strain Sensor Substrate Structure

**DOI:** 10.3390/mi14020355

**Published:** 2023-01-31

**Authors:** Zhenyu He, Wenge Wu, Yunping Cheng, Lijuan Liu

**Affiliations:** School of Mechanical Engineering, North University of China, Taiyuan 030051, China

**Keywords:** substrate structure design, function area, connection area, design criteria

## Abstract

With the intelligent tool cutting force measurement model as the engineering background, the selection, design, and optimization of the substrate structure of the tool-embedded thin-film strain sensor are studied. The structure of the thin-film strain sensor is studied, and the substrate structure design is divided into function area structure design and connection area structure design. Establishing the substrate structure library of the sensor, we subdivide the library into six layouts of function area infrastructure and five layouts of connection area infrastructure. Taking the sensitivity, fatigue life, and comprehensive mechanical properties of the substrate structure as the design indexes, based on the statics theory, the functional relationship between the structural parameters and the deflection of the six layouts of the substrate function area is established; based on the dynamics theory, the functional relationship between the parameters and the natural frequency of six layouts of the function area is established; based on the coupling of structural statics design theory and dynamics design theory, the evaluation method for the comprehensive performance of the parameters of six layouts of the function area is established. Based on the function area structure, five connection area structures are designed for comprehensive performance analysis. The structural sensitivity of the substrate function area design and optimization is expanded 1.75 times, and the comprehensive performance is expanded 1.53 times. The sensitivity of the connection area design and optimization is expanded 2.3 times, and the comprehensive performance is expanded 1.72 times. The structure is optimized according to the structural stress characteristics, the design, selection, and optimization process of the substrate structure summarized herein, and five design criteria of the substrate structure are proposed.

## 1. Introduction

To meet the requirements of intelligent manufacturing and intelligent machining [1,2,3] for real-time monitoring and control [4,5,6,7] of the cutting force, an intelligent tool system is proposed. The thin-film strain force-measuring sensor system is combined with the tool to form an integrated intelligent tool [8,9]. Notably, the thin-film strain force-measuring sensor converts the change of physical parameters of the tool during the cutting process into the monitoring signal and realizes online real-time monitoring of the cutting force. As shown in Figure 1, an intelligent tool system for cutting force measurement is depicted, composed of the tool and the thin-film strain sensor, which is shown in Figure 1a. Likewise, the thin-film strain sensor is embedded with the tool [10,11] to measure the cutting force. As shown in Figure 1b, the thin-film strain sensor [12,13] is composed of the film resistance grid and the elastic substrate, and the elastic substrate is the carrier of the film resistance grid. The tool is subjected to a cutting force, and the deformation of the sensor substrate makes the resistance of the film resistance grid change, causing the output of the measurement circuit to change. The performance of the thin-film strain sensor is determined by the film resistance grid and the substrate, and the sensor’s performance is improved by designing and optimizing the substrate structure and film structure of the thin-film sensor [14,15]. Xuerui Li et al. [16] designed a substrate structure based on the tool handle structure by reducing the thickness of the substrate structure from 1 mm to 0.04 mm, which improved the output voltage of the cutting force measurement. Yunping Cheng et al. [17] also proposed six substrate structures and established the expression of substrate structure parameters and deflection. Likewise, Caiwei Xiao et al. [18] designed a 12 mm substrate, which integrated the sensor and tool, reducing the volume of the measurement system. The problems existing in the design and preparation of thin-film strain sensors are the contradiction between sensitivity and stiffness, the contradiction between sensitivity and natural frequency, and the contradiction between the natural frequency of the sensor system and the vibration interference of the cutting system. Effective design of the substrate structure is the key to the sensor’s measurement accuracy and dynamic characteristics. Additionally, the performance of the substrate structure is of great significance to the mechanical index and stability of the sensor. Sensitivity, high repeatability, and durability are essential for evaluating the thin-film strain sensor [19,20,21]. Thus, the sensor substrate structure’s design will improve its sensitivity, life, and comprehensive performance [22,23].

In this paper, the sensitivity, natural frequency, and comprehensive performance of the thin-film strain sensor substrate structure are taken as the design indexes, the design model of the structural parameters and performance indexes is established, and the comprehensive and complete design theory of the substrate structure is established.

## 2. Design Theory

The principle of the substrate structural design is to open a hole or slot on the substrate stress concentration area to improve the strain performance. The design of the substrate structure can be divided into three design theories based on different performance requirements: (1) The “parameters-deflection *y*” design theory is based on the statics theory, which studies the relationship between structural parameters and deflection. The greater the structural deflection, the better the sensitivity. (2) The “parameters-natural frequency *f*_0_” design theory is based on the dynamics theory, which studies the relationship between structural parameters and natural frequency. The larger the structural natural frequency, the better the fatigue life of the substrate structure. (3) The “parameters-comprehensive performance *Z*” design theory, based on statics theory and dynamics theory coupling, is a comprehensive performance evaluation method of substrate structure proposed, and the substrate structure with good comprehensive performance is selected.

As shown in Figure 1c, the thin-film strain sensor substrate structure, according to functional requirements, can be divided into function area and connection area. The main purpose of the substrate function area is to carry the measurement-sensitive device, improve the strain performance, and better map the cutting force. Additionally, the main function of the substrate connection area of the thin-film strain sensor is to connect and couple with the tool structure. According to the structural division of the substrate, the structural design of the substrate is divided into the substrate function area structural design and the substrate connection area structural design.

The structural design of the substrate function area can be divided into two steps: (1) Establish functional models of structural parameters and different performance indicators. (2) The structure design is based on the main design parameter *B_e_* of the function area structure. As shown in Figure 1c, the parameter *B_e_* is the effective working width of the function area at the substrate length *l*/2 position, which affects the layout and preparation of the film resistance grid. Notably, the structural design of the substrate function area needs to meet *B_e_* ≥ *B*/2. The structural design of the substrate connection area mainly considers the tool structure form, sensor-embedded tool form, mechanical properties, and other aspects. This paper designs and analyzes the connection area structure based on the mechanical properties. Based on different functions and connection area structures, the thin-film strain sensor substrate structure layouts is established, as shown in Figure 2. 

According to the force-measuring characteristics of the embedded tool cutting force measurement sensor, the substrate structure is equivalent to the cantilever beam model for theoretical analysis. As shown in Figure 3, the cantilever beam model of the sensor substrate is established. The basic dimensional parameters of the substrate structure are length *l* = 30 mm, width *B* = 16 mm, and thickness *h* = 0.3 mm; the substrate is AISI 1045 steel, *E* = 210,000 MPa, and *ρ* = 7.8 × 10^3^ kg/m^3^; the mechanical analysis of substrate structure *F* = 1 N, and the relationship between the substrate structure parameters and *y*, *f*_0_, and *Z* is investigated.

## 3. Substrate Function Area Structural Design

The substrate structure library shown in Figure 2 summarizes six substrate function area structures that carry out parameter designs based on statics, dynamics, and comprehensive performance evaluation.

### 3.1. Statics Design

#### 3.1.1. Rectangular Substrate

As shown in Figure 4, the rectangular substrate structure is the most basic structure among sensor substrates and is the template for all substrate structural designs. The rectangular substrate is optimized and improved according to different structural requirements in the application.

The static analysis of the rectangular substrate structure is carried out, and the deflection expression *y* of the substrate along the *x*-axis direction is:(1)$$y=\frac{2F(3lx^{2}-x^{3})}{EBh^{3}}$$
where *F* is the force, *l* is the rectangular substrate length, *h* is the substrate thickness, *B* is the substrate width, *E* is the elastic modulus, and *x* is the coordinate point in the substrate length direction.

The parameters are *F* = 1 N, *B* = 16 mm, *l* = 30 mm, *h* = 0.3 mm, and *E* = 210,000 MPa. Constructing the rectangular substrate structure model for the theoretical calculation and simulation analysis of statics, the resulting deflection is shown in Figure 5 [24,25].

#### 3.1.2. Circular Arc Substrate

The circular arc substrate structure is shown in Figure 6, where the curves on both sides of the function area of the substrate structure is a geometric arc, and the arc radius *r* determines the arc size. The expression of arc curve function *f*(*x*) of the substrate structure is:(2)$$f(x)=\frac{B}{2}+\sqrt{r^{2}-\frac{l^{2}}{4}}-\sqrt{r^{2}-(x-\frac{l}{2}{)}^{2}}$$
where *l* is the circular arc substrate length, *B* is the substrate width, *r* is the arc radius, and *x* is the coordinate point in the substrate length direction.

The static analysis of circular arc substrate structure is carried out, and the deflection expression *y* of the substrate along the *x*-axis direction is:(3)$$y=\frac{3F}{4Eh^{3}}\cdot \left\{\begin{array}{c} \left(2\gamma^{2}-\alpha^{2}-2l\eta \right)\cdot \left(\arcsin\frac{\eta }{\alpha }+\arcsin\frac{l}{\alpha }\right)-\left(\eta +2l\right)\cdot \sqrt{\alpha^{2}-\eta^{2}}+2\gamma \cdot \left(\eta +l\right)+2\gamma \eta \cdot \ln\frac{B}{\gamma -\sqrt{\alpha^{2}-\eta^{2}}}+\left(4x+l\right)\cdot \beta -\\ 2\gamma \cdot \sqrt{\gamma^{2}-\alpha^{2}}\cdot \left\{\begin{array}{c} \arctan\frac{\eta }{\sqrt{\gamma^{2}-\alpha^{2}}}+\arctan\frac{l}{\sqrt{\gamma^{2}-\alpha^{2}}}+\\ \arctan\left[\frac{\gamma }{\sqrt{\gamma^{2}-\alpha^{2}}}\cdot \tan\left(\arcsin\frac{\eta }{\alpha }\right)\right]+\arctan\left[\frac{\gamma }{\sqrt{\gamma^{2}-\alpha^{2}}}\cdot \tan\left(\arcsin\frac{l}{\alpha }\right)\right] \end{array}\right\}+\\ \frac{4l\eta \gamma }{\sqrt{\gamma^{2}-\alpha^{2}}}\cdot \left\{\arctan\left[\sqrt{\frac{\gamma +\alpha }{\gamma -\alpha }}\cdot \tan\left(\frac{1}{2}\arcsin\frac{\eta }{\alpha }\right)\right]+\arctan\left[\sqrt{\frac{\gamma +\alpha }{\gamma -\alpha }}\cdot \tan\cdot \left(\frac{1}{2}\arcsin\frac{l}{\alpha }\right)\right]\right\}+\\ \left\{2l\gamma \cdot \ln\left\{\frac{\left[\gamma -\alpha +\left(\gamma -\alpha \right)\cdot \tan^{2}\left(\frac{1}{2}\arcsin\frac{\eta }{\alpha }\right)\right]\cdot \left[\gamma -\alpha +\left(\gamma +\alpha \right)\cdot \tan^{2}\left(\frac{1}{2}\arcsin\frac{l}{\alpha }\right)\right]}{\left[\gamma -\alpha +\left(\gamma +\alpha \right)\cdot \tan^{2}\left(\frac{1}{2}\arcsin\frac{\eta }{\alpha }\right)\right]\cdot \left[\gamma -\alpha +\left(\gamma -\alpha \right)\cdot \tan^{2}\left(\frac{1}{2}\arcsin\frac{l}{\alpha }\right)\right]}\right\}\right. \end{array}\right\}$$
where
(4)$$\left\{\begin{array}{l} \alpha =2r\\ \beta =\sqrt{4r^{2}-l^{2}}\\ \gamma =B+\beta \\ \eta =2x-l \end{array}\right.$$
where *F* is the force, *h* is the substrate thickness, and *E* is the elastic modulus.

The parameters *F* = 1 N, *B* = 16 mm, *l* = 30 mm, *h* = 0.3 mm, and *E* = 210,000 MPa. The relationship between the arc radius *r* and the free end deflection of the substrate structure is shown in Figure 7.

The arc radius is *r* = 30.125 mm, constructing the circular arc substrate structure model for the theoretical calculation and simulation analysis of statics, and the resulting deflection is shown in Figure 8.

#### 3.1.3. Parabolic Curve Substrate

The parabolic curve substrate structure is shown in Figure 9, and the curves on both side of function area of the substrate structure comprise the geometric parabola. The expression of the arc curve function *f*(*x*) of the substrate structure is:(5)$$f(x)=\frac{2B(n-1)x^{2}}{nl^{2}}-\frac{2B(n-1)x}{nl}+\frac{B}{2}$$
where *l* is the parabolic curve substrate length, *B* is the substrate width, *n* is the size factor, and *x* is the coordinate point in the substrate length direction.

The static analysis of the parabolic curve substrate structure is carried out, and the deflection expression *y* of the substrate along the *x*-axis direction is:(6)$$y=\frac{3nFl^{2}}{2(n-1)EBh^{3}}\cdot \left\{\begin{array}{l} \frac{2(n-1)x-nl}{\sqrt{n-1}}\cdot \arctan(\frac{2\sqrt{n-1}\cdot x}{nl-2(n-1)x})+\\ (\frac{l}{2}-x)\cdot \ln(1+\frac{4(n-1)\cdot (x^{2}-xl)}{nl^{2}})+\\ 2x+\frac{l}{2}\cdot \ln n-\frac{l}{2}\cdot \ln[1+(\frac{2\sqrt{n-1}\cdot x}{l}-\sqrt{n-1}{)}^{2}] \end{array}\right\}$$
where *F* is the force, *h* is the substrate thickness, and *E* is the elastic modulus.

The parameters *F* = 1 N, *B* = 16 mm, *l* = 30 mm, *h* = 0.3 mm, and *E* = 210,000 MPa. The relationship between the size factor *n* and the free end deflection of substrate structure is shown in Figure 10. The size factor is *n* = 2, constructing the parabolic curve substrate structure model for the theoretical calculation and simulation analysis of statics, and the resulting deflection is shown in Figure 11.

#### 3.1.4. Three-Segment Stepped Substrate

As shown in Figure 12, the three-segment stepped substrate structure and its characteristic is that the substrate width *B* changes nonlinearly in the direction of substrate length *l*. The characteristics of the substrate structure parameters are as follows:(7)$$\left\{\begin{array}{l} l_{1}=l_{3}\\ B_{1}/B=\lambda \end{array}\right.$$

The static analysis of the three-segment stepped substrate structure is carried out, and the deflection expression *y* of the substrate along the *x*-axis direction is:(8)$$y=\frac{2F}{Eh^{3}}\cdot \left\{\begin{array}{l} \frac{\left(3l-x)x^{2}\cdot u(x)+[(x+2l_{1})-3l]{\left(x-l_{1}\right)}^{2}\cdot u(x-l_{1}\right)}{B\cdot u(x)}+\\ \frac{\left[3l-(x+2l_{1})]{\left(x-l_{1}\right)}^{2}\cdot u(x-l_{1})+(x-l-2l_{1}){\left(x-l+l_{1}\right)}^{2}\cdot u(x-l+l_{1}\right)}{B\cdot u(x)+B(\lambda -1)\cdot u(x-l_{1})}+\\ \frac{\left(l+2l_{1}-x){\left(x-l+l_{1}\right)}^{2}\cdot u(x-l+l_{1}\right)}{B\cdot u(x)+B(\lambda -1)\cdot u(x-l_{1})+B(1-\lambda )\cdot u(x+l_{1}-l)} \end{array}\right\}$$
where *λ* = *B*_1_/*B*, *B* is the first-segment and third-segment width of the three-segment substrate, *B*_1_ is middle-segment width, *l* is the substrate overall length, *h* is the substrate thickness, *E* is the elastic modulus, *l*_1_ = (*l* − *l*_2_)/2, *l*_2_ is the middle-segment length, *F* is the force, *x* is the coordinate point in the substrate length direction, *u*(*x*) is the unit step function, and the characteristic of the function is:(9)$$u(x)=\left\{\begin{array}{l} 1\;\;\;x\geq 0\\ 0\;\;\;x<0 \end{array}\right.$$

The parameters are *F* = 1 N, *B* = 16 mm, *l* = 30 mm, *h* = 0.3 mm, and *E* = 210,000 MPa. The relationship between the dimensional parameters’ length *l*_1_ and width *B*_1_ of the three-segment stepped substrate and the free end deflection of substrate structure are shown in Figure 13.

The dimensional parameters are *B*_1_ = 8 mm and *l*_1_ = 5 mm, constructing the three-segment stepped substrate structure model for the theoretical calculation and simulation analysis of statics, and the resulting deflection is shown in Figure 14.

#### 3.1.5. Five-Segment Stepped Substrate

As shown in Figure 15, the five-segment stepped substrate structure and the characteristics of the substrate structure parameters are as follows:(10)$$\left\{\begin{array}{l} l_{1}=l_{2}=l_{4}=l_{5}\\ B_{1}/B=\lambda_{1}\\ B_{2}/B=\lambda_{2} \end{array}\right.$$

The width change of the five-segment stepped substrate is expressed by function *B*(*x*):(11)$$B(x)=\left\{\begin{array}{l} B\cdot u(x)+(\lambda_{1}-1)B\cdot u(x-l_{1})+(\lambda_{2}-\lambda_{1})B\cdot u(x-2l_{1})+\\ \left(\lambda_{1}-\lambda_{2})B\cdot u(x+2l_{1}-l)+(1-\lambda_{1})B\cdot u(x+l_{1}-l\right) \end{array}\right\}$$

The static analysis of the five-segment stepped substrate structure is carried out, and the deflection expression *y* of the substrate along the *x*-axis direction is:(12)$$y=\sum_{i=1}^{5}\frac{2F}{Eh^{3}B(x)}\left\{\begin{array}{l} {}[3l-(x+2\sum_{m=0}^{i-1}l_{m})](x-\sum_{m=0}^{i-1}l_{m}{)}^{2}\cdot [u(x-\sum_{m=0}^{i-1}l_{m})-u(x-\sum_{m=0}^{i}l_{m})]+\\ 2l_{i}(x-\sum_{m=0}^{i-1}l_{m})[3(l-\sum_{m=0}^{i-1}l_{m})-2l_{i}]\cdot u(x-\sum_{m=0}^{i}l_{m})+l_{i}{}^{2}[(x+2\sum_{m=0}^{i}l_{m})-3l]\cdot u(x-\sum_{m=0}^{i}l_{m}) \end{array}\right\}$$
where *λ* = *B*_1_/*B*, *λ* = *B*_2_/*B*, *B* is the first-segment and fifth-segment width of five-segment substrate, *B*_1_ is the second-segment and fourth-segment width of five-segment substrate, *B*_2_ is the third-segment width, *l* is the substrate overall length, *h* is the substrate thickness, *E* is the elastic modulus, *l*_1_ = *l*_2_, *l*_1_ is the first-segment and fifth-segment length of five-segment substrate, *l*_2_ is the second-segment and fourth-segment length of five-segment substrate, *F* is the force, *x* is the coordinate point in the substrate length direction, and *u*(*x*) is the unit step function.

The parameters are *F* = 1 N, *B* = 16 mm, *l* = 30 mm, *l*_1_ = 5 mm, *h* = 0.3 mm, and *E* = 210,000 MPa. The relationship between the dimensional parameters’ width *B*_1_ and width *B*_2_ of the five-segment stepped substrate and the free end deflection of substrate structure is shown in Figure 16.

The dimensional parameters are *B*_1_ = 12 mm and *B*_2_ = 8 mm, constructing the five-segment stepped substrate structure model for the theoretical calculation and simulation analysis of statics, and the resulting deflection is shown in Figure 17.

#### 3.1.6. Opening Hole Substrate

Opening hole substrate structure is shown in Figure 18, and its characteristic is opening at the center of the substrate function area. The characteristics of the substrate structure parameters are as follows:(13)$$\left\{\begin{array}{l} l_{1}=l_{3}\\ B_{1}/B=\lambda \end{array}\right.$$

The static analysis of the opening hole substrate structure is carried out, and the deflection expression *y* of the substrate along the *x*-axis direction is:(14)$$y=\frac{2F}{Eh^{3}}\cdot \left\{\begin{array}{l} \frac{\left(3l-x)x^{2}\cdot u(x)+[(x+2l_{1})-3l]{\left(x-l_{1}\right)}^{2}\cdot u(x-l_{1}\right)}{B\cdot u(x)}+\\ \frac{\left[3l-(x+2l_{1})]{\left(x-l_{1}\right)}^{2}\cdot u(x-l_{1})+(x-l-2l_{3}){\left(x-l+l_{3}\right)}^{2}\cdot u(x-l+l_{3}\right)}{B\cdot u(x)-\lambda B\cdot u(x-l_{1})}+\\ \frac{\left(l+2l_{3}-x){\left(x-l+l_{3}\right)}^{2}\cdot u(x-l+l_{3}\right)}{B\cdot u(x)-\lambda B\cdot u(x-l_{1})+\lambda B\cdot u(x-l_{1}-l_{2})} \end{array}\right\}-\frac{6M}{Eh^{3}}\cdot \left\{\begin{array}{l} \frac{x^{2}\cdot u(x)-{\left(x-l_{1}\right)}^{2}\cdot u(x-l_{1})}{2B}+\\ \frac{{\left(x-l_{1}-l_{2}\right)}^{2}\cdot u(x-l_{1}-l_{2})}{2B} \end{array}\right\}$$
where *M* = *FB*_1_/8, *λ* = *B*_1_/*B*, *B* is the opening hole substrate width, *B*_1_ is the opening hole width, *l* is the substrate overall length, *h* is the substrate thickness, *E* is the elastic modulus, *l*_1_ = (*l* − *l*_2_)/2, *l*_2_ is the opening hole length, *F* is the force, *x* is the coordinate point in the substrate length direction, and *u*(*x*) is the unit step function.

The parameters *F* = 1 N, *B* = 16 mm, *l* = 30 mm, *l*_1_ = 5 mm, *h* = 0.3 mm, and *E* = 210,000 MPa. The relationship between the opening width *B*_1_ of the opening hole substrate and the free end deflection of the substrate structure is shown in Figure 19.

The dimensional parameter is *B*_1_ = 8 mm, constructing the opening hole substrate structure model for the theoretical calculation and simulation analysis of statics, and the resulting deflection are shown in Figure 20.

Among the six layouts of the function area, the sensitivity of the three-segment stepped substrate is better than others, and the statics theoretical calculation results are basically consistent with the simulation results.

### 3.2. Dynamics Design

The dynamic analysis of substrate structure is carried out to study the relationship between the structural parameters and the natural frequency *f*_0_. The following is the parameter design based on dynamic analysis for six kinds of substrate function area structures.

#### 3.2.1. Rectangular Substrate

The natural frequency *f*_0_ theory design of rectangular substrate structure is:(15)$$f_{0}=\frac{0.172h}{l^{2}}\sqrt{\frac{E}{\rho }}$$
where *l* is the rectangular substrate length, *h* is the substrate thickness, *E* is the elastic modulus, and *ρ* is the density. The parameters are *h* = 0.3 mm, *E* = 210,000 MPa, and *ρ* = 7.8 × 10^3^ kg/m^3^. The relationship between the length *l* and the natural frequency of the substrate structure is shown in Figure 21a. The dimensional parameter is *l* = 30 mm, constructing the rectangular substrate structure model for the theoretical calculation and simulation analysis of dynamics, and the natural frequency is shown in Figure 21b.

#### 3.2.2. Circular Arc Substrate

The natural frequency *f*_0_ theory design of circular arc substrate structure is:(16)$$f_{0}=\frac{1.155h}{\pi }\sqrt{\frac{E}{\left\{\begin{array}{l} \rho \left\{Bl+\frac{\beta l}{2}-\frac{\alpha^{2}}{2}\cdot \arcsin(\frac{l}{\alpha })\right\}\cdot \\ \left\{\begin{array}{l} (4\gamma^{2}-2\alpha^{2}-4l^{2})\cdot (\arcsin\frac{l}{\alpha })+\frac{8l^{2}\gamma }{\sqrt{\gamma^{2}-\alpha^{2}}}\cdot \left\{\arctan\left[\sqrt{\frac{\gamma +\alpha }{\gamma -\alpha }}\cdot \tan(\frac{1}{2}\arcsin\frac{l}{\alpha })\right]\right\}+\\ 4l\gamma +2l\beta -4\gamma \cdot \sqrt{\gamma^{2}-\alpha^{2}}\cdot \left\{\arctan\frac{l}{\sqrt{\gamma^{2}-\alpha^{2}}}+\arctan\left[\frac{\gamma }{\sqrt{\gamma^{2}-\alpha^{2}}}\cdot \tan(\arcsin\frac{l}{\alpha })\right]\right\} \end{array}\right\} \end{array}\right\}}}$$
where *α*, *β*, *γ* are shown in Formula (4); *B* is the circular arc substrate width; *l* is the substrate length; *h* is the substrate thickness; *E* is the elastic modulus; and *ρ* is the density.

The parameters are *B* = 16 mm, *l* = 30 mm, *h* = 0.3 mm, *E* = 210,000 MPa, and *ρ* = 7.8 × 10^3^ kg/m^3^. The relationship between the arc radius *r* and the natural frequency of the substrate structure is shown in Figure 22a. The arc radius is *r* = 30.125 mm, constructing the circular arc substrate structure model for the theoretical calculation and simulation analysis of dynamics, and the natural frequency is shown in Figure 22b.

#### 3.2.3. Parabolic Curve Substrate

The natural frequency *f*_0_ theory design of parabolic curve substrate structure is:(17)$$f_{0}=\frac{1.414h}{\pi l^{2}}\sqrt{\frac{(n-1)lE}{(n+2)\rho \cdot \left[\frac{(n-2)l}{\sqrt{n-1}}\cdot \arctan(\frac{2l\sqrt{n-1}}{(2-n)l})+2l\right]}}$$
where *l* is the parabolic curve substrate length, *n* is the size factor, *h* is the substrate thickness, *E* is the elastic modulus, and *ρ* is the density.

The parameters are *l* = 30 mm, *h* = 0.3 mm, *E* = 210,000 MPa, and *ρ* = 7.8 × 10^3^ kg/m^3^. The relationship between the size factor *n* and the natural frequency of substrate structure is shown in Figure 23a. The size factor is *n* = 2, constructing the parabolic curve substrate structure model for the theoretical calculation and simulation analysis of dynamics, and the natural frequency is shown in Figure 23b.

#### 3.2.4. Three-Segment Stepped Substrate

The natural frequency *f*_0_ theory design of a three-segment stepped substrate structure is:(18)$$f_{0}=\frac{0.5h}{\pi }\sqrt{\frac{\lambda EB}{\rho \left\{\lambda Bl+2(1-\lambda )Bl_{1}\right\}\cdot \left\{(1-\lambda )[3ll_{1}{}^{2}-3l^{2}l_{1}-2l_{1}{}^{3}]+l^{3}\right\}}}$$
where *λ* = *B*_1_/*B*, *B* is the first-segment and third-segment width of three-segment substrate, *B*_1_ is the middle-segment width, *l* is the substrate overall length, *h* is the substrate thickness, *E* is the elastic modulus, *ρ* is the density, *l*_1_ = (*l* − *l*_2_)/2, and *l*_2_ is the middle-segment length.

The parameters are *l*_1_ = 5 mm, *l* = 30 mm, *B* = 16 mm, *h* = 0.3 mm, *E* = 210,000 MPa, and *ρ* = 7.8 × 10^3^ kg/m^3^. The relationship between the width *B*_1_ and the natural frequency of substrate structure is shown in Figure 24a. The dimensional parameter is *B*_1_ = 8 mm, constructing the three-segment stepped substrate structure model for the theoretical calculation and simulation analysis of dynamics, and the natural frequency is shown in Figure 24b.

#### 3.2.5. Five-Segment Stepped Substrate

The natural frequency *f*_0_ theory design of a five-segment stepped substrate structure is:(19)$$f_{{}_{0}}=\frac{0.5h}{\pi }\sqrt{\frac{\lambda_{1}\lambda_{2}EB}{\rho \left\{\begin{array}{l} \left[\lambda_{2}Bl+2Bl_{1}\cdot (1+\lambda_{1}-2\lambda_{2})\right]\cdot \\ \left[\begin{array}{l} 3l^{2}l_{1}\cdot (\lambda_{1}\lambda_{2}+\lambda_{2}-2\lambda_{1})-3ll_{1}{}^{2}\cdot (\lambda_{1}\lambda_{2}+3\lambda_{2}-4\lambda_{1})+\\ 2l_{1}{}^{3}\cdot (\lambda_{1}\lambda_{2}+7\lambda_{2}-8\lambda_{1})+\lambda_{1}l^{3} \end{array}\right] \end{array}\right\}}}$$
where *λ* = *B*_1_/*B*, *λ* = *B*_2_/*B*, *B* is the first-segment and fifth-segment width of five-segment substrate, *B*_1_ is the second-segment and fourth-segment width of five-segment substrate, *B*_2_ is the third-segment width, *l* is the substrate overall length, *h* is the substrate thickness, *E* is the elastic modulus, *ρ* is the density, *l*_1_ = *l*_2_, *l*_1_ is the first-segment and fifth-segment length of five-segment substrate, and *l*_2_ is the second-segment and fourth-segment length of five-segment substrate.

The parameters are *l*_1_ = 5 mm, *l* = 30 mm, *B* = 16 mm, *h* = 0.3 mm, *E* = 210,000 MPa, and *ρ* = 7.8 × 10^3^ kg/m^3^. The relationship between the width *B*_1_, width *B*_2_, and the natural frequency of substrate structure is shown in Figure 25a. The dimensional parameters are *B*_1_ = 12 mm and *B*_2_ = 8 mm, constructing the five-segment stepped substrate structure model for the theoretical calculation and simulation analysis of dynamics, and the natural frequency is shown in Figure 25b.

#### 3.2.6. Opening Hole Substrate

The natural frequency *f*_0_ theory design of opening hole substrate structure is:(20)$$f_{0}=\frac{2h}{\pi }\sqrt{\frac{(1-\lambda )BE}{\rho \left[(1-\lambda )Bl+2\lambda Bl_{1}\right]\cdot \left\{16\left[\lambda (3ll_{1}{}^{2}-3l^{2}l_{1}-2l_{1}{}^{3})+l^{3}\right]-3\lambda (1-\lambda )Bll_{1}\right\}}}$$
where *λ* = *B*_1_/*B*, *B* is the opening hole substrate width, *B*_1_ is the opening hole width, *l* is the substrate overall length, *h* is the substrate thickness, *E* is the elastic modulus, *ρ* is the density, *l*_1_ = (*l* − *l*_2_)/2, and *l*_2_ is the opening hole length.

The parameters are *l*_1_ = 5 mm, *l* = 30 mm, *B* = 16 mm, *h* = 0.3 mm, *E* = 210,000 MPa, and *ρ* = 7.8 × 10^3^ kg/m^3^. The relationship between the width *B*_1_ and the natural frequency of the substrate structure is shown in Figure 26a. The dimensional parameter is *B*_1_ = 8 mm, constructing the opening hole substrate structure model for the theoretical calculation and simulation analysis of dynamics, and the natural frequency is shown in Figure 26b.

Among the six layouts of the function area, the natural frequency of the rectangular substrate is better than others, and the dynamics theoretical calculation results are basically consistent with the simulation results.

### 3.3. Comprehensive Analysis of Substrate Structure

Six kinds of substrate structures are constructed for the theoretical calculation and simulation analysis of statics and dynamics. Among the six kinds of structures, the effective width at substrate function area *l*/2 position of (2), (3), (4), (5), and (6) substrate structures are equal, that is, *B_e_* = 8 mm. By comparing the statics theoretical calculation results and simulation results of the six kinds of structures, the deflection values of the six structures are shown in Figure 27 and arranged from large to small: (4) (6) (2) (3) (5) (1). By comparing the dynamics theoretical calculation results and simulation results of the six kinds of structures, the natural frequency values of the six structures are shown in Figure 28 and arranged from large to small: (1) (3) (2) (5) (6) (4).

Substrate structure based on statics design, the greater the value of *y*(*l*), the better the sensitivity of the substrate. Substrate structure based on dynamics design, the greater the value of *f*_0_, the better the performance of the substrate. However, when the sensitivity *y*(*l*) increases, the natural frequency *f*_0_ decrease. When designing the substrate structure, the sensitivity and natural frequency of the structure should be comprehensively weighed and analyzed. As shown in formula 21, the statics and dynamics design coupling model of the substrate structure is established, and *Z* is the design quality factor of the structure. Notably, the greater the value of *Z*, the better the comprehensive performance of the substrate structure.
(21)$$Z=y(l)\cdot f_{0}$$

In the comprehensive performance analysis of six kinds of substrate structures, as shown in Figure 29, the quality factor *Z* of the three-segment stepped substrate structure (4) is the largest, and the comprehensive performance is the best. Compared with the original rectangular substrate structure, the sensitivity of the three-segment stepped substrate structure is expanded 1.75 times, and the comprehensive performance is expanded 1.53 times.

Select the (4) substrate structure for designing and optimizing the substrate connection area structure.

## 4. Substrate Connection Area Structural Design

The substrate structure library shown in Figure 2 summarizes five kinds of substrate connection area structures to carry out parameter design based on statics, dynamics, and comprehensive performance evaluation.

It can be seen from the comprehensive performance analysis of the substrate function area structures that the comprehensive performance index *Z* of the three-segment stepped substrate is the highest among the six kinds of substrate function area structures. Based on the three-segment stepped substrate, five kinds of connection area structures, as shown in Figure 30, are designed for mechanical analysis.

Analyze the design parameters of the five kinds of substrate connection area structures, among which (1), (2), (3), and (5) had equal groove areas of the connection zone, and in (4) and (5), the groove size of the connection zone was equal.

### 4.1. Statics Structural Analysis

The statics analysis of the five kinds of connection structures is shown in Figure 31a~e, and the deflection values of the five kinds of structures shown in Figure 31f are arranged from large to small: (5) (3) (1) (2) (4).

### 4.2. Dynamics Structural Analysis

The dynamics analysis of the five kinds of connection structures is shown in Figure 32a~e, and the natural frequency values of the five kinds of structures shown in Figure 32f are arranged from large to small: (4) (2) (1) (3) (5).

### 4.3. Comprehensive Analysis and Optimization of the Substrate Structure

In the comprehensive performance analysis of the five kinds of substrate structures, as shown in Figure 33, the quality factor *Z* of the connection area structure (5) is the largest, and the comprehensive performance was the best. Compared with the original rectangular substrate structure, the structure’s sensitivity (5) is expanded 2.3 times, and the comprehensive performance is expanded 1.72 times. Select the structure (5) to prepare the substrate, as shown in Figure 34.

Considering that the right angle of the transition part between the connection area and function area of the substrate is prone to stress concentration, the concentrated stress will weaken the stiffness of the local structure of the substrate and damage the quality of the film preparation on the substrate surface. To reduce the influence of stress concentration on the sensor’s structural and mechanical properties and thin-film preparation [24], the transition part’s right angle is optimized to a round angle, as shown in Figure 35.

To study the influence of substrate structure on film preparation [26,27], as shown in Figure 36, the film prepared on the surface of three-segment stepped substrate structure (a) and rectangular substrate structure (b) has an evident and wide range of cracks, and the cracks spread from the substrate connection area to the center of the substrate function area along the direction of the red arrow, resulting in film damage and quality degradation. The crack range and diffusion trend of the optimized structure (c) are better than those of the two structures (a) and (b).

Based on the above design theories and optimization method, five design criteria for substrate structure design are proposed: (a) Based on the size and layout of the sensor resistance grid, to set and design the constraint conditions of the function area structure parameter, select the appropriate substrate function area structure. (b) Based on the tool structure form, sensor embedded tool form, mechanical properties, and others, design and select the appropriate substrate connection area structure. (c) Under the condition of material safety, based on the statics structural analysis theory, design and select the substrate structure with good deflection performance. (d) Considering the working characteristics of the machine tool processing system, tool cutting force system, and thin-film sensor system, and based on the dynamics structural analysis theory, design and select the substrate structure that meets the working frequency requirements. (f) A comprehensive evaluation of the statics and dynamic performance of the substrate structure according to the weight of different performance requirements is conducted to design, select, and optimize the substrate structure that meets the comprehensive requirements.

## 5. Conclusions

In this paper, the design theory and optimization method of the substrate structure of the tool-embedded thin-film strain sensor are studied, and the influence of the structure parameters on the sensitivity, fatigue life, and comprehensive performance of the substrate is studied. Establishing the substrate structure library divides the substrate structure design into function and connection area structure design. Based on statics, design theoretical models between the six kinds of substrate function area structure parameters and deflection *y* are established; based on dynamics, design theoretical models between the six kinds of structure parameters and natural frequency *f*_0_ are established; the comprehensive evaluation design method of coupling the statics design theory and dynamics design theory of the structural parameters of the function area is established, and the comprehensive performance evaluation quality factor *Z* is proposed. Based on the engineering requirements of the function area structure parameter *B_e_*, six kinds of function area structure models are constructed for theoretical calculation and simulation calculation. It is concluded that the three-segment stepped substrate structure has the best comprehensive performance. Based on the three-segment stepped substrate structure, five kinds of substrate connection area structure schemes are designed and the models are constructed. Through statics simulation analysis, dynamics simulation analysis, and comprehensive performance quality factor *Z* analysis for the five models, it is concluded that the comprehensive performance of the structural scheme (5) is the best. Compared with the original rectangular structure, the three-segment stepped structure has 1.75 times more sensitivity and 1.53 times more comprehensive performance. The connection area structure has 2.3 times more sensitivity and 1.72 times more comprehensive performance. Therefore, we summarized the design, selection, and optimization process of substrate structure and put forward five substrate structure design criteria.

## Figures and Tables

**Figure 1 micromachines-14-00355-f001:**
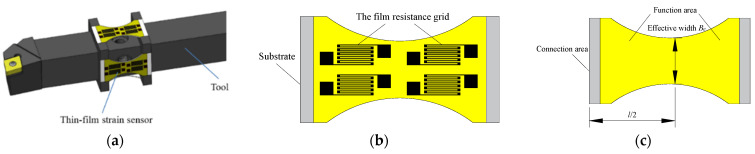
Tool-embedded thin-film strain sensor cutting force measurement system. (**a**) Intelligent tool. (**b**) Sensor structure. (**c**) Substrate structure.

**Figure 2 micromachines-14-00355-f002:**
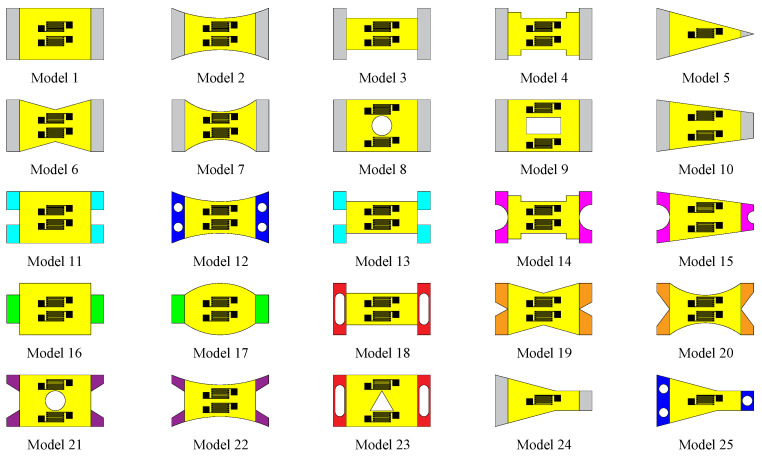
Substrate structure layouts.

**Figure 3 micromachines-14-00355-f003:**
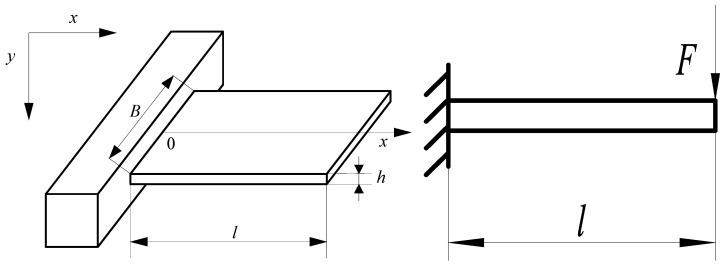
Analysis model.

**Figure 4 micromachines-14-00355-f004:**
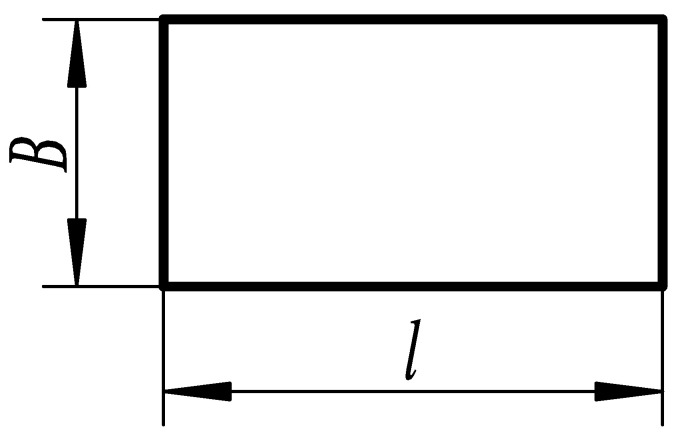
Rectangular substrate structure layout.

**Figure 5 micromachines-14-00355-f005:**
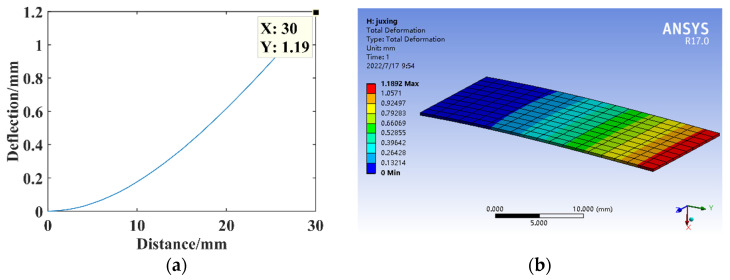
Static analysis of rectangular structure. (**a**) Deflection calculation. (**b**) Deflection simulation.

**Figure 6 micromachines-14-00355-f006:**
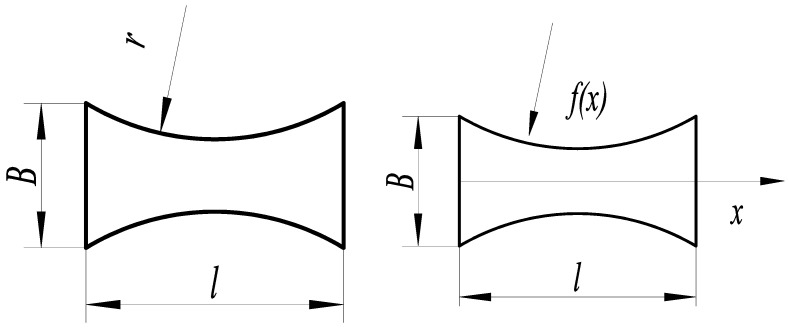
Circular arc structure.

**Figure 7 micromachines-14-00355-f007:**
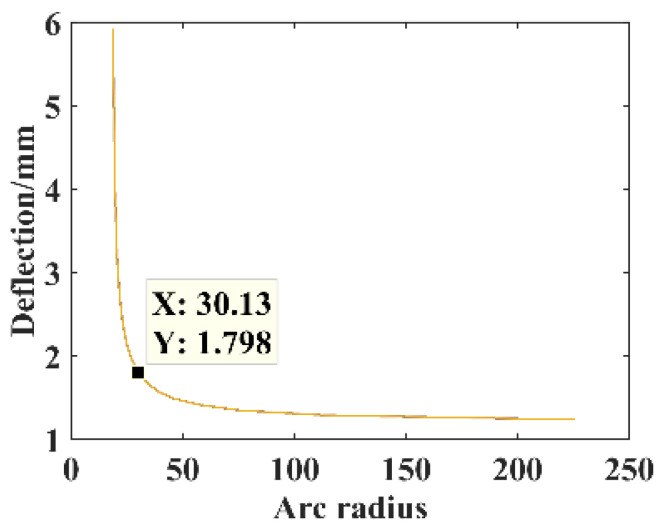
Relationship between the parameter *r* and deflection.

**Figure 8 micromachines-14-00355-f008:**
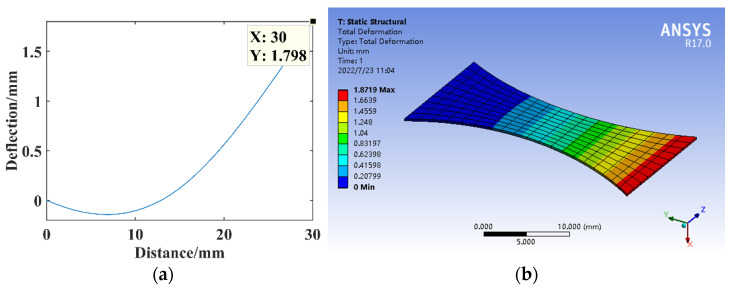
Statics analysis of circular arc structure. (**a**) Deflection calculation. (**b**) Deflection simulation.

**Figure 9 micromachines-14-00355-f009:**
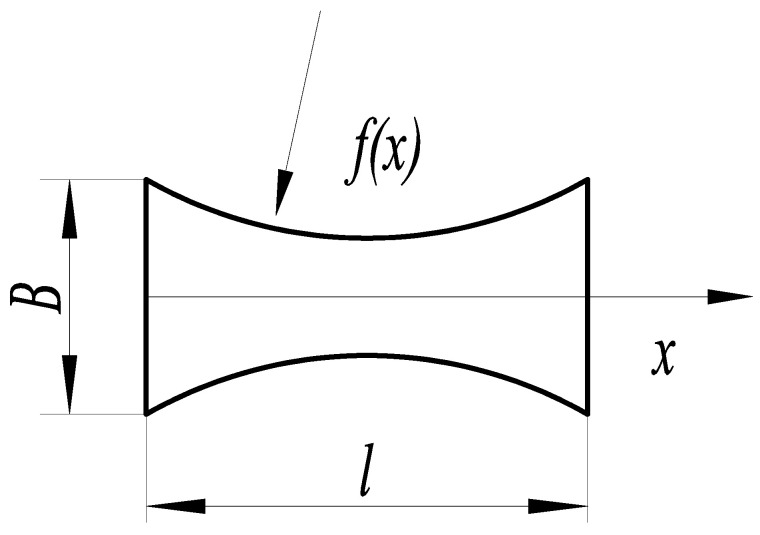
Parabolic curve structure.

**Figure 10 micromachines-14-00355-f010:**
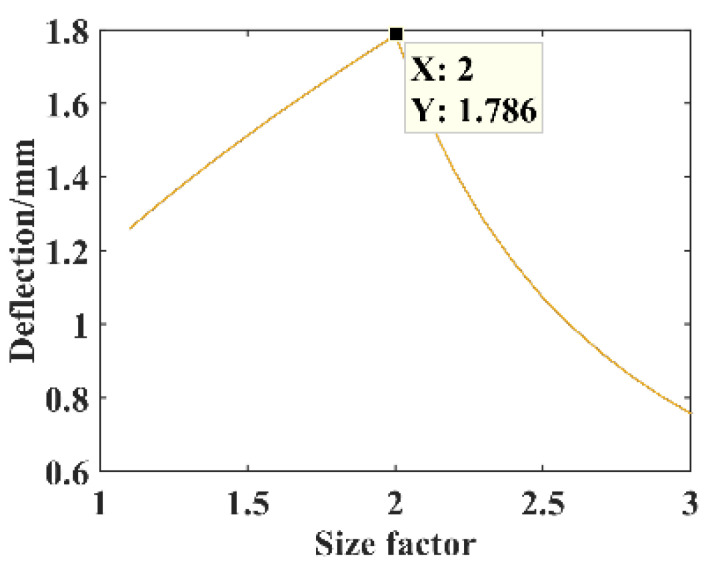
Relationship between the size factor *n* and deflection.

**Figure 11 micromachines-14-00355-f011:**
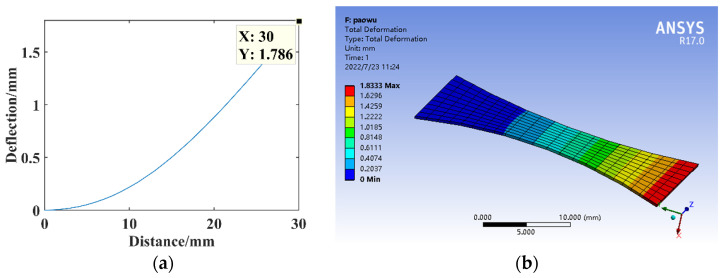
Statics analysis of parabolic curve structure. (**a**) Deflection calculation. (**b**) Deflection simulation.

**Figure 12 micromachines-14-00355-f012:**
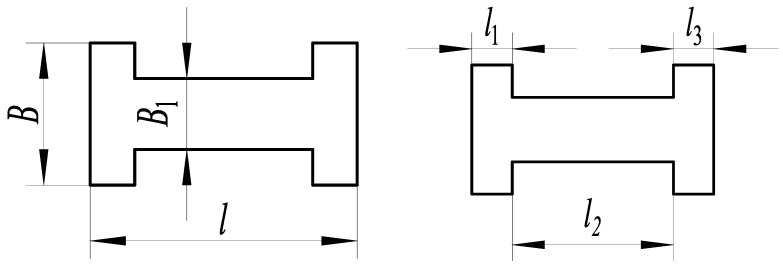
Three-segment stepped structure.

**Figure 13 micromachines-14-00355-f013:**
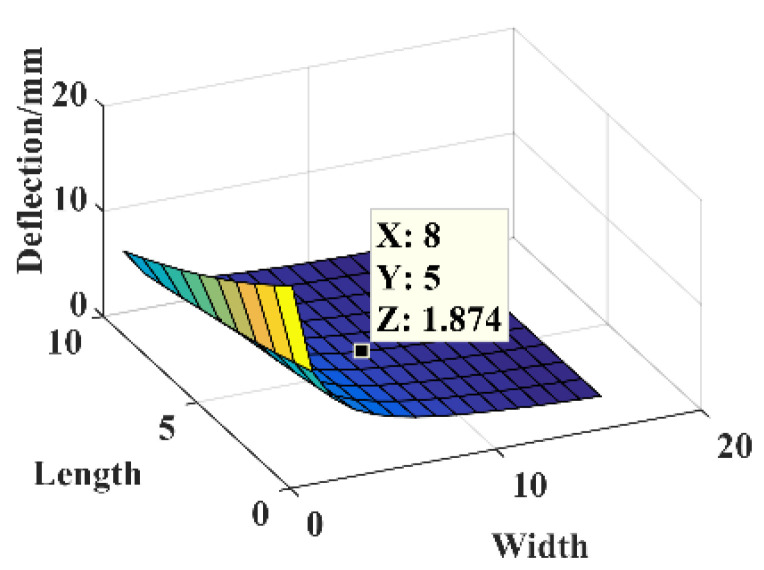
Relationship between the parameters *B*_1_, *l*_1_, and deflection.

**Figure 14 micromachines-14-00355-f014:**
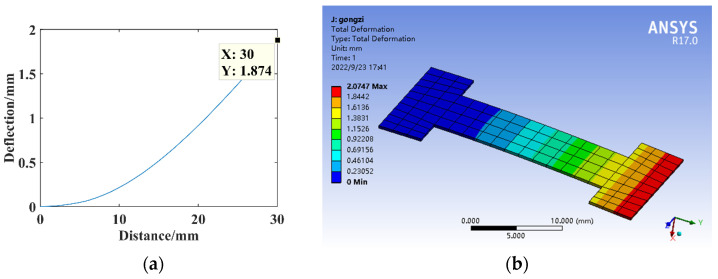
Statics analysis of three-segment stepped structure. (**a**) Deflection calculation. (**b**) Deflection simulation.

**Figure 15 micromachines-14-00355-f015:**
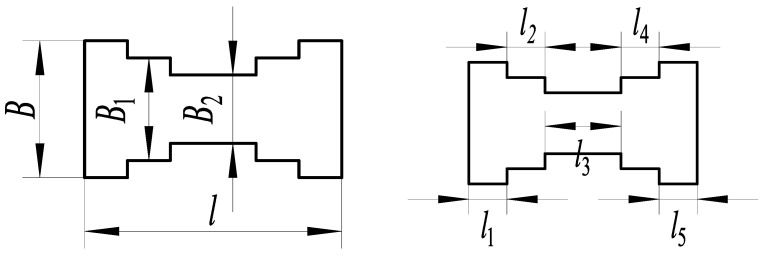
Five-segment stepped structure.

**Figure 16 micromachines-14-00355-f016:**
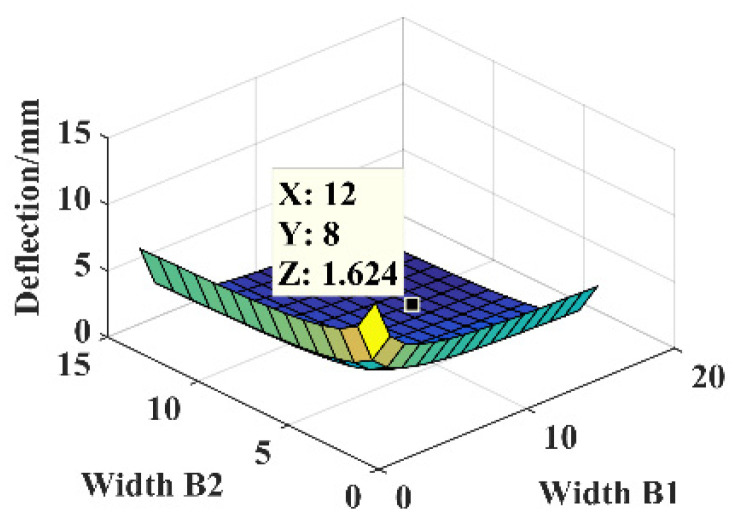
Relationship between the parameters *B*_1_, *B*_2_, and deflection.

**Figure 17 micromachines-14-00355-f017:**
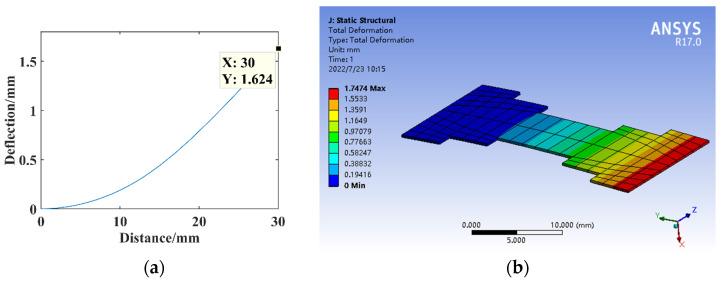
Statics analysis of five-segment stepped structure. (**a**) Deflection calculation. (**b**) Deflection simulation.

**Figure 18 micromachines-14-00355-f018:**
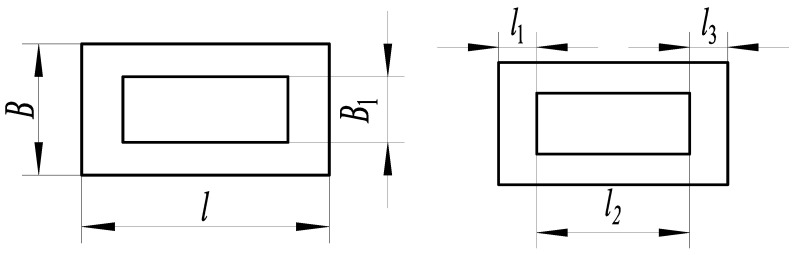
Opening hole structure.

**Figure 19 micromachines-14-00355-f019:**
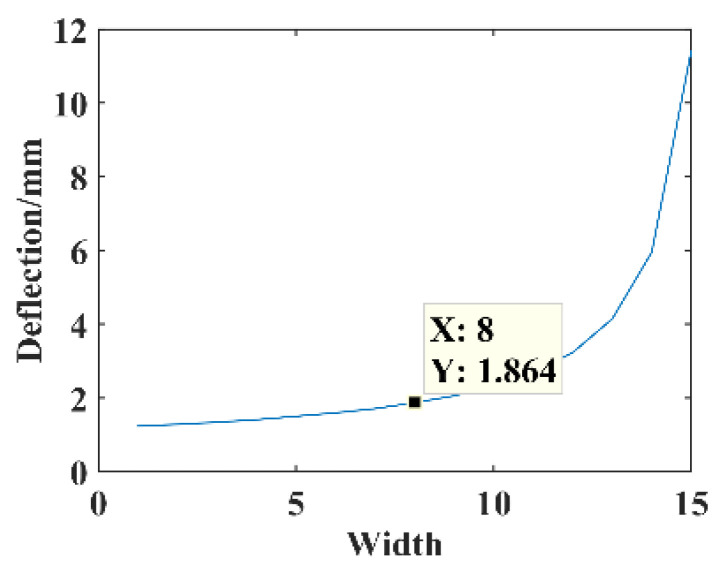
Relationship between the parameter *B*_1_ and deflection.

**Figure 20 micromachines-14-00355-f020:**
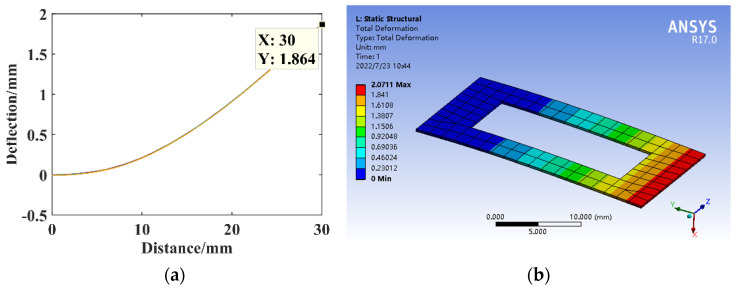
Statics analysis of opening hole structure. (**a**) Deflection calculation. (**b**) Deflection simulation.

**Figure 21 micromachines-14-00355-f021:**
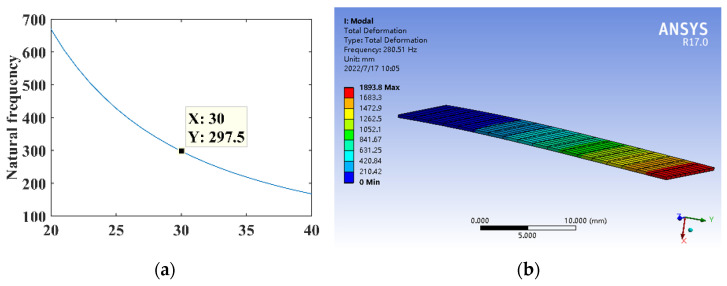
Dynamics analysis of rectangular substrate structure. (**a**) Natural frequency theory design. (**b**) Dynamic analysis.

**Figure 22 micromachines-14-00355-f022:**
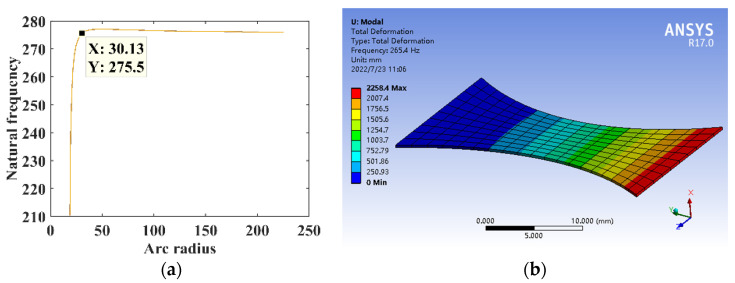
Dynamics analysis of circular arc substrate structure. (**a**) Natural frequency theory design. (**b**) Dynamic analysis.

**Figure 23 micromachines-14-00355-f023:**
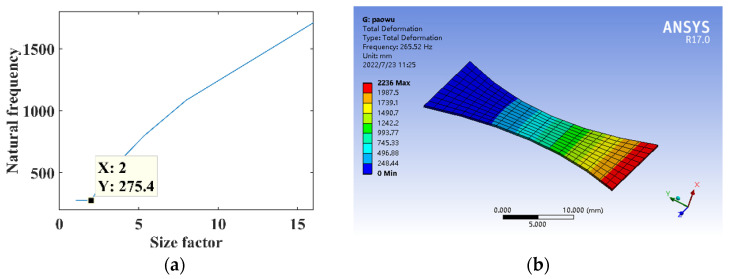
Dynamics analysis of parabolic curve substrate structure. (**a**) Natural frequency theory design. (**b**) Dynamic analysis.

**Figure 24 micromachines-14-00355-f024:**
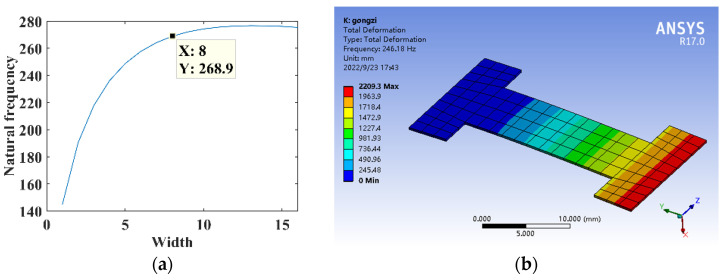
Dynamics analysis of three-segment stepped substrate structure. (**a**) Natural frequency theory design. (**b**) Dynamic analysis.

**Figure 25 micromachines-14-00355-f025:**
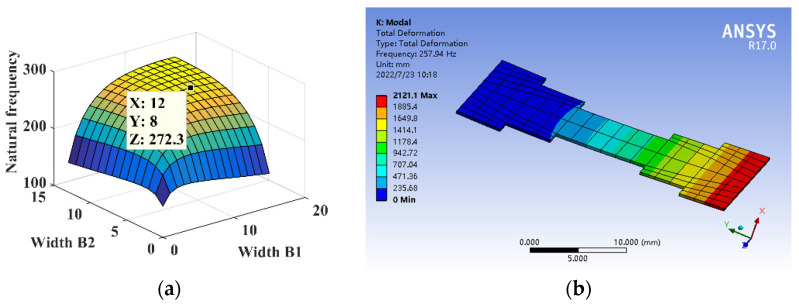
Dynamics analysis of five-segment stepped substrate structure. (**a**) Natural frequency theory design. (**b**) Dynamic analysis.

**Figure 26 micromachines-14-00355-f026:**
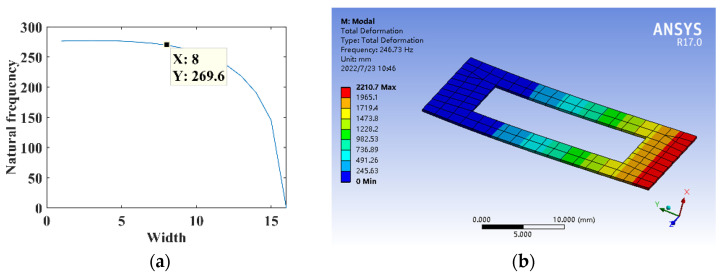
Dynamics analysis of opening hole substrate structure. (**a**) Natural frequency theory design. (**b**) Dynamic analysis.

**Figure 27 micromachines-14-00355-f027:**
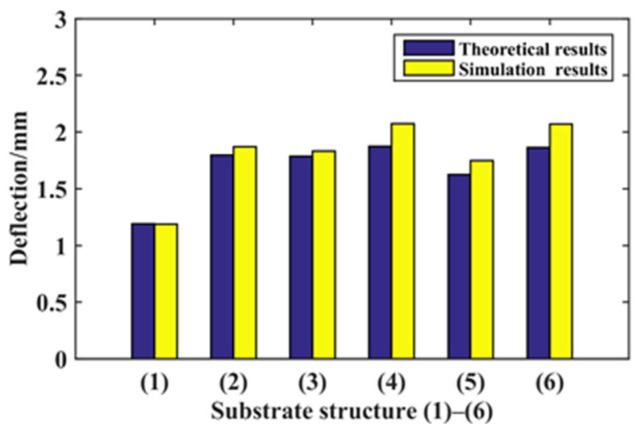
Six structures’ statics analysis.

**Figure 28 micromachines-14-00355-f028:**
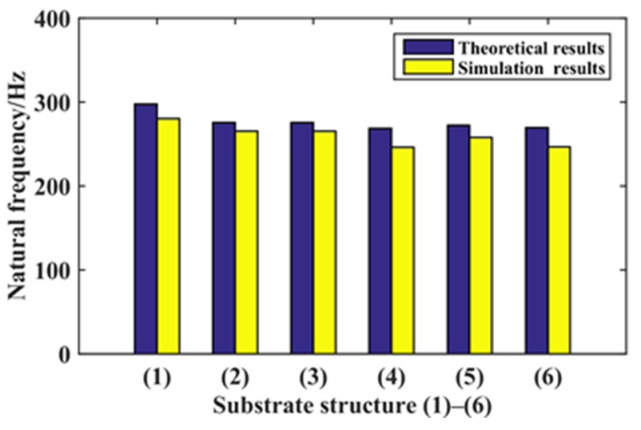
Six structures’ dynamics analysis.

**Figure 29 micromachines-14-00355-f029:**
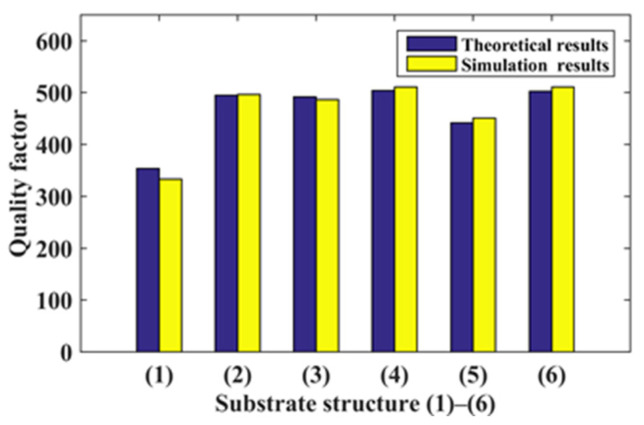
Six structures’ quality factor.

**Figure 30 micromachines-14-00355-f030:**
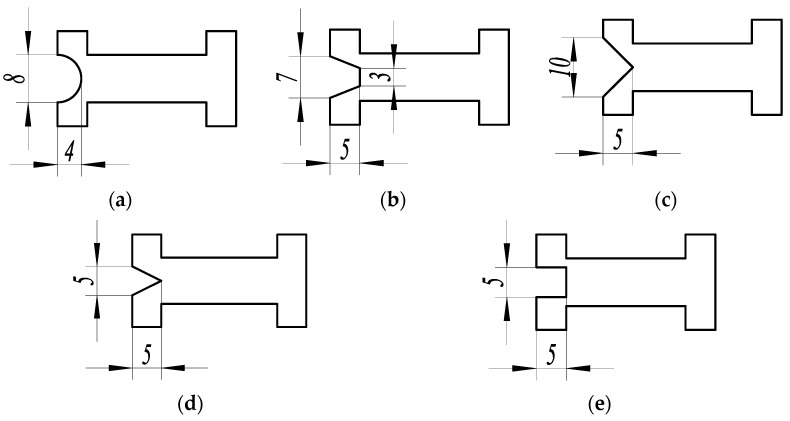
Five kinds of substrate connection area structure schemes. (**a**) Structure scheme (1), (**b**) Structure scheme (2), (**c**) Structure scheme (3), (**d**) Structure scheme (4), and (**e**) Structure scheme (5).

**Figure 31 micromachines-14-00355-f031:**
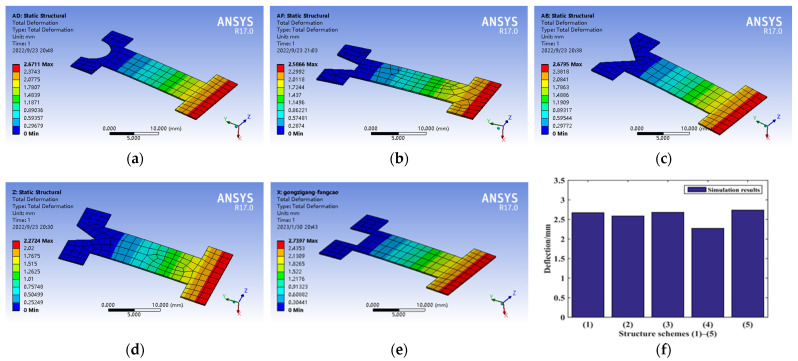
Statics performance analysis of connection structure. (**a**) Structure scheme (1), (**b**) Structure scheme (2), (**c**) Structure scheme (3), (**d**) Structure scheme (4), (**e**) Structure scheme (5), and (**f**) Structure scheme comparison.

**Figure 32 micromachines-14-00355-f032:**
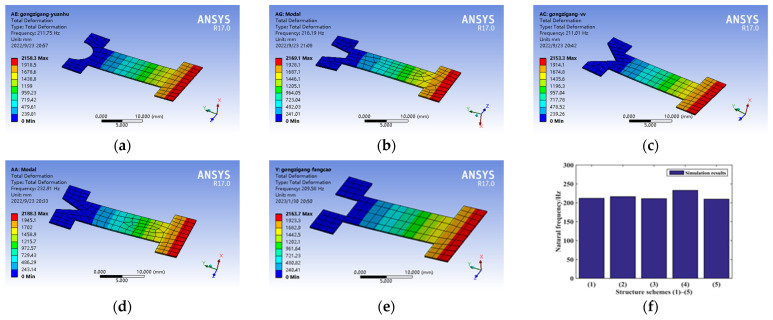
Dynamics performance analysis of connection structure. (**a**) Structure scheme (1), (**b**) Structure scheme (2), (**c**) Structure scheme (3), (**d**) Structure scheme (4), (**e**) Structure scheme (5), and (**f**) Structure scheme comparison.

**Figure 33 micromachines-14-00355-f033:**
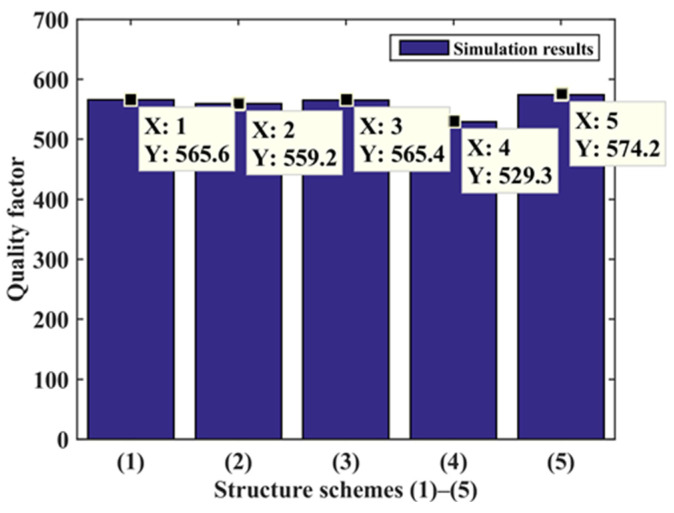
Quality factor.

**Figure 34 micromachines-14-00355-f034:**
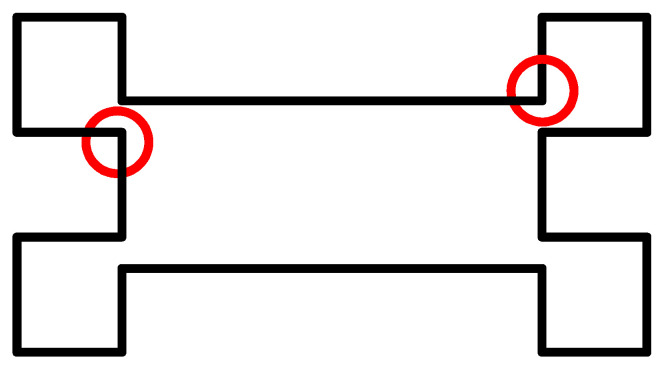
Substrate structural design.

**Figure 35 micromachines-14-00355-f035:**
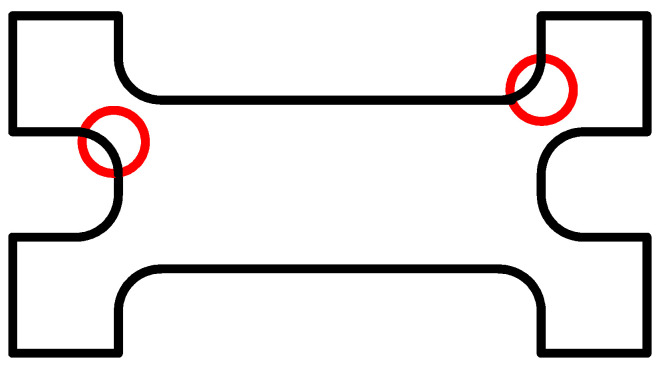
Substrate structural optimization.

**Figure 36 micromachines-14-00355-f036:**
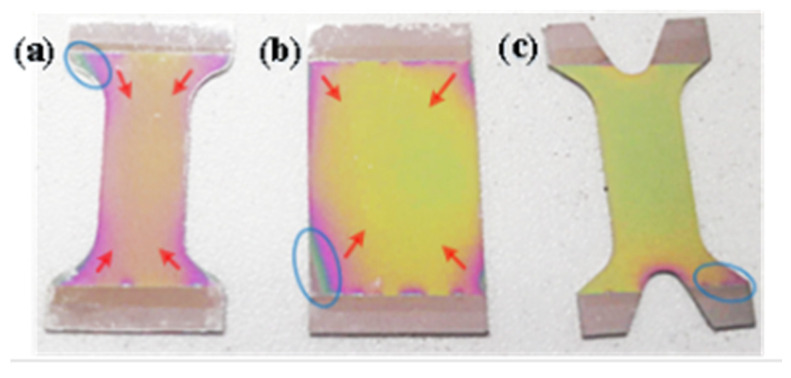
The relationship between the substrate structure and film quality [27].

## Data Availability

The data used to support the findings of this study are available from the corresponding author upon request.

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
