# Peer review of "Design and Optimization of Tool-Embedded Thin-Film Strain Sensor Substrate Structure"

_micromachines, 2023, doi:10.3390/mi14020355_

Round 1

Author Response

Response to Reviewer 1 Comments

  1. In the entire paper, the author has explained deflection function and curve function. But where from the author has taken the equation either from any reference or the author’s proof?

Answers: The equations are original, for example the equation (3) derivation process is shown in follow:

The curve expression y″ of the circular arc substrate along the x-axis direction is: 

where

The slope expression y′ of the circular arc substrate along the x-axis direction is: 

when x=0, y′=0

The deflection expression y of the circular arc substrate along the x-axis direction is: 

when x=0, y=0

The equation (3) is:

  1. Here, the comprehensive performance has been chosen an optimization parameter but in the explanation of influence of substrate structure of film quality, author has explained about cracks propagation resulting in film damage. But author has explained as ‘Z’ factor, so, how crack optimization has been explained with ‘Z’ factor? Could the author explain how ‘Z’ factor is associated with crack propagation?

Answers: The factor ‘Z’ is the comprehensive performance evaluation index of the statics and dynamics performance of the substrate structure, which is independent of the crack. This paper mentions that the crack propagation is related to the local stress concentration of the substrate structure. The greater the local stress concentration, the worse the film quality. In this paper, by optimizing the local structure of the substrate, the local stress concentration is reduced and the film quality is improved.

  1. In the case of natural frequency theory design, how frequency is associated with crack propagation in substrate design? Does it vary or will it be constant?

Answers: The crack propagation is related to the local stress concentration of the substrate structure and independent of the natural frequency. In the chapter 2 "Substrate structure design theory and method", the "parameters-natural frequency f0" design theory based on the dynamic theory, which studies the relationship between structural parameters and natural frequency. The larger the structural natural frequency, the better the fatigue life of the substrate structure. The research on the relationship between frequency and crack propagation is not involved in this paper, but this is a good idea. I will conduct more in-depth research around this idea in the future.

  1. 4.In case of dynamics structural design, how stress is affected with frequency design? In case of comprehensive analysis and optimization of substrate structure, how come the structure schemes of 3 and 5 as shown in figure 32, are of same natural frequency and why structure of scheme 4, has higher frequency?

Answers: In case of dynamics structural design, the greater the frequency, the smaller the stress. As shown in Figure 30, the slotted area of the substrate of Scheme 3 and Scheme 5 is equal, that is 50mm2, and the slotted area of Scheme 4 is 25mm2 that is minimum. The smaller the slotted area, the smaller the stress of the substrate structure, and the greater the frequency. So, the structure schemes of 3 and 5 as shown in figure 32 are same natural frequency, and the structure of scheme 4, has higher frequency.

Reviewer 2 Report

The Manuscript deals with the design and optimization of a thin film strain sensor embedded in tool. Stress is made on substrate sensor design. Complete sensor features are presented under consistent theory formulation. Results Presented are consistent. The subject is relevant and the work worth to be published. However the manuscript presents several technical faulty concerning insufficient explanation of models and standard scientific English writing, compromising normal readability and scientific understanding. To many short paragraphs, telegraphic writing style meaningless sentence mission prepositions. One can find also miss a consistent discussion on results. Under this compliance I would recommend the manuscript to be fully revised and resubmitted.

Author Response

Response to Reviewer 2 Comments

  1. The Manuscript deals with the design and optimization of a thin film strain sensor embedded in tool. Stress is made on substrate sensor design. Complete sensor features are presented under consistent theory formulation. Results Presented are consistent. The subject is relevant and the work worth to be published. However the manuscript presents several technical faulty concerning insufficient explanation of models and standard scientific English writing, compromising normal readability and scientific understanding. To many short paragraphs, telegraphic writing style meaningless sentence mission prepositions. One can find also miss a consistent discussion on results. Under this compliance I would recommend the manuscript to be fully revised and resubmitted.

Answers: For the comments and suggestions, we made fully revise to the English style and technical problem of the manuscript, and the revisions to the resubmit manuscript is marked up using the "Track", please review please review the attachment.

Round 2

Reviewer 1 Report

the author has answered properly which can be accepted for publication.

Author Response

Response to Reviewer 1 Comments

Dear editors and reviewers:

Thank you for your rigorous academic attitude.

Sincerely yours,

Wenge Wu

Reviewer 2 Report

The manuscript deals with the design and optimization of a thin film strain sensor embedded in a tool. Stress is made on substrate sensor design. Complete sensor features are presented under consistent theory formulation. Presented results are consistent. The subject is relevant and worth to be included in a scientific work. This is a revised version of a previous one, in which authors carried out extensive corrections which significantly improved the manuscript.

However, there are still several technical issues concerning legends and equations labeling, insufficient explanation of used models and modeling software used ant its assumptions and methods. Abstract is also too long with many short paragraphs, reconsider rewriting. Comparisons of deflection and resonance curves for different substract proposed layouts. As Authors supplied a word version of revised manuscript, I took the liberty of including all may detailed comments on it, which I am attaching.

Author Response

Response to Reviewer 2 Comments

Dear editors and reviewers:

Firstly, we would like to thank you for your kind letter and the reviewers’ constructive comments concerning our article (micromachines-2108513). These comments are all valuable and helpful for improving our article. All the authors have seriously discussed all these comments. According to the reviewers’ comments, we have tried our best to modify our manuscript to meet the requirements of your journal. For the comments and suggestions, the revisions to the resubmit manuscript is marked up using the "Track". In this revised version, changes to our manuscript within the document were all highlighted by using red-colored text. Thank you again for your rigorous academic attitude.

Sincerely yours,

Wenge Wu